# Pressurized Hot Liquid Extraction with 15% *v/v* Glycerol-Water as An Effective Environment-Friendly Process to Obtain *Durvillaea incurvata* and *Lessonia spicata* Phlorotannin Extracts with Antioxidant and Antihyperglycemic Potential

**DOI:** 10.3390/antiox10071105

**Published:** 2021-07-10

**Authors:** Fernanda Erpel, María Salomé Mariotti-Celis, Javier Parada, Franco Pedreschi, José Ricardo Pérez-Correa

**Affiliations:** 1Chemical and Bioprocess Engineering Department, School of Engineering, Pontificia Universidad Católica de Chile, Santiago 7820436, Chile; faerpel@uc.cl (F.E.); fpedreschi@ing.puc.cl (F.P.); 2Escuela de Nutrición y Dietética, Universidad Finis Terrae, Santiago 7501015, Chile; mmariotti@uft.cl; 3Institute of Food Science and Technology, Faculty of Agricultural and Food Sciences, Universidad Austral de Chile, Valdivia 5110566, Chile; javier.parada@uach.cl

**Keywords:** brown seaweeds, phlorotannins, pressurized hot liquid extraction, glycerol-water mixtures, environment-friendly, food-grade, antioxidant activity, carbohydrate-hydrolyzing enzymes, mannitol, heavy metals

## Abstract

Brown seaweed phlorotannins have shown the potential to promote several health benefits. *Durvillaea incurvata* and *Lessonia spicata*—species that are widely distributed in central and southern Chile—were investigated to obtain phlorotannin extracts with antioxidant and antihyperglycemic potential. The use of an environmentally friendly and food-grade glycerol-based pressurized hot liquid extraction (PHLE) process (15% *v/v* glycerol water) was assessed for the first time to obtain phlorotannins. Multiple effects were analyzed, including the effect of the species, harvesting area (Las Cruces and Niebla), and anatomical part (holdfast, stipe, and frond) on the extracts’ polyphenol content (TPC), antioxidant capacity (AC), and carbohydrate-hydrolyzing enzyme—α-glucosidase and α-amylase—inhibitory activity. Contaminants, such as mannitol, heavy metals (As, Cd, Pb, Hg, and Sn), and 5-hydroxymethylfurfural (HMF), were also determined. The anatomical part used demonstrated a significant impact on the extracts’ TPC and AC, with holdfasts showing the highest values (TPC: 95 ± 24 mg phloroglucinol equivalents/g dry extract; DPPH: 400 ± 140 μmol Trolox equivalents/g dry extract; ORAC: 560 ± 130 μmol TE/g dry extract). Accordingly, holdfast extracts presented the most potent α-glucosidase inhibition, with *D. incurvata* from Niebla showing an activity equivalent to fifteen times that of acarbose. Only one frond and stipe extract showed significant α-glucosidase inhibitory capacity. No α-amylase inhibition was found in any extract. Although no HMF was detected, potentially hazardous cadmium levels (over the French limit) and substantial mannitol concentrations—reaching up to 50% of the extract dry weight—were found in most seaweed samples and extracts. Therefore, further purification steps are suggested if food or pharmaceutical applications are intended for the seaweed PHLE extracts obtained in this study.

## 1. Introduction

Seaweeds, or marine macroalgae, are a valuable source of nutrients, such as amino acids, dietary fiber (e.g., fucoidan and carrageenan), vitamins (A, B, C, D, and E), minerals (e.g., Ca, P, and I), and essential fatty acids (ω-3 and ω-6) [1]. Southern Chilean coastal populations (Monte Verde, 14,600 years b.p.) included them as part of their diet and medicines [2], similarly to eastern Asian cultures, which have been consuming them for hundreds of years [3]. In the Occident, seaweeds are mainly used to produce gelling or thickening agents for the food and pharmaceutical industries (e.g., carrageenan and alginates); however, their use as “functional foods” is growing, given the accumulating evidence in the literature for their potential health-promoting effects, such as antidiabetic [4], anticancer [5], and antibacterial [6] effects. Nevertheless, since some species accumulate heavy metals and possess high iodine concentrations, there is increasing concern about algae consumption’s potential health risks [7].

Seaweeds are classified into three types: Chlorophyta (green algae), Phaeophyta (brown algae), and Rhodophyta (red algae). Brown algae have attracted greater interest due to their high content of bioactive compounds [8]. Phlorotannins, a group of polyphenols only found in Phaeophyta, stand out due to their superior antioxidant capacity and valuable biological activities [9,10]. Phlorotannins are phloroglucinol (1,3,5-trihydroxy benzene) polymers with chain or net-like structures [11]. They are located in cell vesicles called physodes, in a soluble form and are strongly associated with the cell wall’s proteins and alginates [12]. Phlorotannins’ primary function is to protect seaweeds from stress; hence, their concentrations vary according to abiotic and biotic factors (e.g., UV radiation, temperature, and herbivores) [13]. Antidiabetic capacity is one of the most frequently explored bioactivities of phlorotannins. Phlorotannin-rich extracts and isolated phlorotannins (e.g., dieckol) from algae of the genera *Ecklonia* and *Ishige* have shown antihyperglycemic potential similar to that of acarbose, a commercial antidiabetic drug [14,15]. This response is mainly attributed to their capacity to inhibit the carbohydrate-hydrolyzing enzymes of the gut—α-glucosidase and α-amylase—and to modulate crucial enzymes in glucose metabolism in the liver and muscles [16,17].

Different extraction processes have been tested to obtain phlorotannin-rich extracts with potential food and pharmaceutical applications. As phlorotannins are moderately polar compounds, high efficiencies have been achieved using maceration with acetone, ethanol, or solvent-water mixtures. However, maceration involves high solvent volumes and long extraction times [10,18]. Centrifugal partition extraction with 50:50 ethyl acetate-water has also shown high efficiency, with increased productivity [19]. Phlorotannin-rich extracts have also been prepared through environmentally friendly extraction techniques, such as microwave-assisted extraction, ultrasound-assisted extraction, and pressurized hot liquid extraction (PHLE) [19,20].

PHLE with green solvents has been postulated as the most favorable environmentally friendly method for extracting various compounds from plants and algae [21]. PHLE involves applying solvents at temperatures higher than their boiling point under high pressure (1500 psi) to keep them in a liquid state. These conditions enhance the analyte’s solubility and mass transfer rate, thus reducing extraction times and solvent consumption. Additionally, as PHLE reduces the dielectric constant of water, extractions with water or hydroalcoholic mixtures (e.g., water-glycerol) effectively extract polar and moderately polar compounds. PHLE with hydroalcoholic mixtures has been successfully applied to obtain food-grade polyphenols from plant and seaweed matrices [21,22]. However, the harsh operating conditions of PHLE might cause the co-extraction of undesirable compounds, such as heavy metals and reducing sugars, and the neoformation of potential human carcinogens, such as 5-hydroxymethylfurfural (HMF) [23,24]. Therefore, PHLE extracts could not only present technological problems but also pose a risk for consumers’ health.

This study aimed to obtain phlorotannin-rich extracts with antioxidant and antihyperglycemic potential from seaweeds that are broadly distributed in central and southern Chile—*Durvillaea incurvata* and *Lessonia spicata*—using PHLE with 15% *v/v* glycerol-water as a food-grade and environment-friendly extraction approach [25]. The effects of sepcies, harvesting area, and the anatomical part used (i.e., holdfast, stipe, or frond) on the extracts’ polyphenol content, antioxidant capacity, and carbohydrate-hydrolyzing enzyme inhibitory activity were assessed. In our previous PHLE study, 15% *v/v* glycerol-water was shown to be the best solvent, compared with 15% *v/v* ethanol-water and water, to maximize the polyphenol recovery from grape pomace [26]. PHLE with ethanol-water mixtures has been extensively employed to obtain polyphenols from seaweed [19,20,27]; to the best of our knowledge, this study is the first to report on the performance of a glycerol-based PHLE process for this purpose. The current study also raises awareness about the potentially toxic heavy metal content of seaweed extracts through evidencing cadmium levels higher than those permitted by French legislation in some phlorotannin extracts. Thus, the characterization presented here provides further information for the effective design of purification processes intended to remove such contaminants.

## 2. Materials and Methods

### 2.1. Chemicals and Reagents

Most chemicals were acquired from Merck, Germany, including (a) analytical grade solvents: glycerol, acetone, and methanol; (b) HPLC-grade acetonitrile; and (c) analytical grade reagents: Folin-Ciocalteu phenol reagent, 85% ortho-phosphoric acid, glacial acetic acid, NaOH, soluble starch, and the salts KH_2_PO_4_, K_2_HPO_4_, K_4_[Fe(CN)_6_]⋅3H_2_O, ZnSO_4_⋅7H_2_O, KNaC_4_H_4_O6⋅4H_2_O, and NaCl. Other materials were purchased from Sigma-Aldrich, USA, including (a) 2,2-diphenyl-1-picrylhydrazyl (DPPH), fluorescein, 2,2’-Azobis(2-amidinopropane) dihydrochloride (AAPH), 3,5-dinitrosalicylic acid, and p-Nitrophenyl-α-D-glucopyranoside (PNPG); (b) the enzymes α-glucosidase from *S. cerevisiae* (type I, G5003-100UN) and α-amylase from porcine pancreas (type VI-B, A3176-5MU); and (c) the standards phloroglucinol, Trolox, mannitol, and HMF.

Dinitrosalicylic acid color reagent was prepared by dissolving 5 g of 3,5-dinitrosalicylic acid in 400 mL of deionized water. Then, 150 g of potassium sodium tartrate tetrahydrate (KNaC_4_H_4_O_6_⋅4H_2_O) and 8 g of NaOH were slowly added under agitation at 70 °C. The solution was made up to 500 mL with deionized water.

### 2.2. Seaweed Samples

Five replicates of *D. incurvata* (Suhr) Macaya [28]—ex *D. antarctica*—and five replicates of *L. spicata* (Suhr) Santelices [29]—ex *L. nigrescens*—were collected in spring (November 2018) from Las Cruces (33°30′09″ S 71°37′59″ W), Valparaíso, and Niebla (39°52′12″ S 73°23′55″ O), Valdivia, Chile. They were morphologically identified by considering taxonomical characteristics, such as thallus shape, the number of pyrenoids, the presence/absence of marginal teeth, and the cross-section thickness.

The samples were rinsed with tap water and deionized water to remove epiphytes, sand, and salt. They were dried with absorbent paper and split into the three thallus parts: holdfast, stipe, and frond (blades). Each seaweed part was cut into small pieces and frozen at −80 °C until freeze-drying (FDT-8620, Operon, Gimpo, Korea). Finally, the dried samples were milled and stored at −20 °C before the extractions. The twelve studied groups (five replicates each) are shown in Table 1.

### 2.3. Liquid Extraction Methods: PHLE and Maceration with 60% Acetone

All samples were extracted using PHLE. Additionally, one randomly selected sample per species/thallus part combination was extracted by maceration with 60% acetone used as a reference [10]. The PHLE process effectivity was determined through a comparison against the acetone control, considering the polyphenol content and the extracts’ antioxidant capacity. The mean solid:liquid ratio of both extraction procedures was 1:12 *w/v*. Some extractions were carried out in triplicate in both methods to confirm low variability (CV < 10%).

For PHLE, 10 g of ground sample was mixed with (i) 60 g of quartz sand or (ii) 20 g of diatomaceous earth (stipe and frond of *D. incurvata*) and poured into a 100-mL stainless steel extraction cell previously filled with 45 g of quartz sand. The extraction was performed in an accelerated solvent extractor (Dionex ASE 150, Thermo Fisher Scientific, Waltham, MA, USA) at 150 °C and 1500 psi, using a 15% *v/v* glycerol-water solution as a solvent; these operational conditions maximized the recovery of grape pomace polyphenols in a previous optimization study [26]. Two extraction cycles of 5 min were sequentially applied, followed by rinsing with 100 mL of solvent and purging with pressurized nitrogen (total extraction time: 20 min). For maceration, 1 g of sample was extracted with 12 mL of 60% *v/v* acetone-water solution at 30 °C for 1 h in an orbital shaker incubator (ZWYR-240, LABWIT, Melbourne, Australia). The extract was filtered; deionized water was added to reach 50 mL of solution. All extracts were stored at −20 °C until analysis.

### 2.4. Total Solids Content of the PHLE Extracts

A gravimetric method was applied to determine the dry matter content of the PHLE extracts. About 1 g (0.9 mL) of the extract was evenly distributed in the tared dish of an infrared moisture analyzer (MS-70, A&D Weighing, San Jose, CA, USA) and allowed to dry at 150 °C until reaching a constant weight (15–20 min). The final weight corresponded to the dry matter content in 0.9 mL of extract. Measurements were performed in triplicate. Total solids were used to calculate the extraction yields on a dry extract basis (% *w/w*, g of extracted solids per g dry seaweed).

### 2.5. Total Polyphenol Content

Phlorotannin content was estimated using the Folin–Ciocalteu method [30]. In brief, 0.5 mL of extract, standard, or extraction solvent (blank), along with 0.25 mL of Folin–Ciocalteu reagent (diluted 1:2 with deionized water), was sequentially transferred to 3.75 mL of deionized water. The solution was homogenized, and 0.5 mL of 10% *w/v* Na_2_CO_3_ was added. The solution was mixed and left to stand in the dark for 1 h at room temperature. The absorbances were measured at 750 nm (Genesys 150 UV-Vis, Thermo Fisher Scientific, Waltham, MA, USA) and compared to a phloroglucinol calibration curve (15–150 mg/L). All analyses were carried out in duplicate. The results were expressed as mg of phloroglucinol equivalents (mg PE) per g of dry seaweed and g of dry extract.

### 2.6. Antioxidant Capacity 

#### 2.6.1. DPPH Radical Scavenging Activity (DPPH RSA)

The DPPH method was performed as described by other authors [31,32], with modifications. First, a 0.238 mg/mL DPPH stock was prepared in methanol and maintained at 4 °C. Before each batch, a working solution (0.048 mg/mL) was prepared by diluting the DPPH stock (1 in 5). For the analysis, 0.5 mL of the DPPH working solution was added to 0.5 mL of methanol diluted extract or standard. The solution was homogenized and left in the dark for 30 min at room temperature. Sample blanks were prepared by mixing 0.5 mL of the extract with 0.5 mL of methanol to eliminate the extract color interference. The absorbances were read at 520 nm (Genesys 150 UV-Vis, Thermo Fisher Scientific, Waltham, MA, USA) and compared with a Trolox curve (2.5–8 mg/L). All analyses were performed in duplicate. Results were expressed as μmol of Trolox equivalents (μmol TE) per g of dry seaweed and g of dry extract.

#### 2.6.2. Oxygen Radical Absorbance Capacity (ORAC)

ORAC analysis was carried out as reported previously [33], with some modifications. In a 96-well microplate, 250 μL of 55 nM fluorescein prepared in 75 mM phosphate buffer pH 7.4 was added to 25 μL of extract or standard (diluted in the buffer). The microplate was incubated for 30 min at 37 °C, and the reaction was initiated by adding 25 μL of freshly prepared 153 mM AAPH solution (in the phosphate buffer). Fluorescence (excitation: 485 nm; emission: 528 nm) was recorded every 1 min for 1 h at 37 °C in a Synergy HTX multi-mode microplate reader (Biotek Instruments, Winooski, VT, USA). All analyses were performed in triplicate. Gen5 Data Analysis Software (Biotek Instruments, Winooski, VT, USA) was used to determine the area under the curve (AUC) of the samples, standards, and phosphate buffer blanks (AUC _blank_). The net AUC values of samples or standards were calculated as follows:(1)$$\;\mathrm{Net}\;\mathrm{AUC}={\mathrm{AUC}}_{\mathrm{sample}\;\mathrm{or}\;\mathrm{Std}.}-{\mathrm{AUC}}_{\mathrm{blank}}$$

ORAC values of the samples were interpolated from the curve Net AUC Std. vs. Trolox Concentration (2–10 mg/L) and were expressed as μmol of Trolox equivalents (μmol TE) per g of dry seaweed and g of dry extract. 

### 2.7. Inhibition of Carbohydrate-Hydrolyzing Enzymes

#### 2.7.1. Sample Preparation 

Equivolumetric mixtures of the five replicates per sample group were prepared and freeze-dried (FreeZone 4.5 L -50 Freeze Dryer, Labconco, Kansas City, MO, USA). Stock assay solutions (10 mg/mL) were prepared by re-suspending the dried samples in each enzyme buffer. Both inhibition analyses were performed in duplicate, as previously reported [34].

#### 2.7.2. Inhibition of α-Glucosidase Activity 

First, sample dilutions (10–4500 μg/mL), PNPG (5 mM), and α-glucosidase from *S. cerevisiae* (0.1 U/mL) were prepared in 100 mM phosphate buffer pH 6.9. In a 96-well microplate, equal volumes (50 μL) of each sample dilution and 5mM PNPG were mixed and incubated at 37 °C for 5 min. The reaction was initiated by adding 100 μL of 0.1 U/mL α-glucosidase. The absorbances (405 nm) were recorded every 3 min for 9 min at 37 °C (Gemini XPS, Molecular Devices, San Jose, CA, USA). The commercial antidiabetic drug acarbose was also assayed as a positive control. Enzyme activities (%) were determined by subtracting sample blank readings (no enzyme, Abs _blank_) from reaction absorbances (Abs _sample_), and results were compared to the control (no sample, Abs _control_) as follows:(2)$$\;\%\mathrm{Activity}=\frac{{\mathrm{Abs}}_{\mathrm{sample}\;}-{\mathrm{Abs}}_{\mathrm{blank}}\;}{{\mathrm{Abs}}_{\mathrm{control}}}\times 100$$

IC_50_ values (μg/mL) were also determined as the extract concentration that produced 50% inhibition of the enzyme.

#### 2.7.3. Inhibition of α-Amylase Activity 

Prior to the analysis, sample dilutions (10–4500 μg/mL), starch solution (0.5% *w/v*), and porcine pancreatic α-amylase (0.5 mg/mL) were prepared in 20 mM phosphate buffer pH 6.9 (with 6 mM sodium chloride). Then, equal volumes (100 μL) of each sample dilution and 0.5% starch were mixed and incubated at 25 °C for 10 min in a water bath. An aliquot of 100 μL of 0.5 mg/mL α-amylase was added to the mixtures and incubated again at 25 °C for 10 min. The reactions were stopped by adding 200 μL of dinitrosalicylic acid color reagent and incubating at 100 °C for 5 min. Once room temperature was reached, 50 μL of the samples were transferred to a 96-well microplate and diluted with 200 μL of deionized water. The absorbances were read at 540 nm (Gemini XPS, Molecular Devices, San Jose, CA, USA). The commercial antidiabetic drug acarbose was also assayed as the positive control. Enzyme activities (%) were calculated using equation 2.

### 2.8. Contaminant Content

#### 2.8.1. Mannitol

Mannitol was quantified using high-performance liquid chromatography coupled with refractive index detection (HPLC-RI). In brief, 850 μL of deionized water was added to 650 μL of extract and homogenized with a vortex. Then, 3.5 mL of acetonitrile was added to the diluted extract and vortexed again. The mixture was centrifuged for 15 min at 6000 rpm and 20 °C (MIKRO 220R, Hettich, Kirchlengern, Germany) and passed through a 0.22-μm Nylon syringe filter. Then, 20 μL of the sample was injected into an HPLC-IR system (Dionex Ultimate 3000, Thermo Fisher Scientific, Waltham, MA, USA), equipped with a standard phase amino column (LiChroCART^®^ 250-4,6 Purospher^®^ STAR NH₂ (5 µm), Merck, Darmstadt, Germany). Chromatographic separation was carried out at 1 mL/min and 50 °C for 18 min, using 80:20 *v/v* acetonitrile/250 mM H_3_PO_4_ in water as the mobile phase. Under these operating conditions, mannitol eluted at 7.4 min. Analyses were performed in duplicate, and results were expressed as mg of mannitol per g of dry seaweed and g of dry extract.

Before the analyses, the method was validated for linearity, specificity, accuracy, and precision [35]. Linearity was assessed in the range of 100–1000 mg/L and expressed by the coefficient of determination (Table 2). The limit of detection (LOD) and limit of quantification (LOQ) were determined using the calibration curve, according to the following equations:(3)                                  LOD=(3.3×RSD)/ b
(4) LOQ=(10×RSD)/ b
where RSD is the residual standard deviation of the linear regression and *b* is the slope of the regression line.

#### 2.8.2. Hydroxymethylfurfural 

HMF was determined through HPLC coupled with diode array detection (HPLC-DAD), based on a previously described method [36], with modifications. In summary, 1 mL of 1:99 *v/v* acetic acid in deionized water was added to 1 mL of extract and homogenized. Then, 300 μL of carrez I (15% *w/v* K_4_[Fe(CN)_6_]⋅3H_2_O in water) and 300 μL of carrez II (30% *w/v* ZnSO_4_⋅7H_2_O in water) were poured into the solution. The mixture was vortexed for 30 s and centrifugated for 15 min at 6000 rpm and 20 °C (MIKRO 220R, Hettich, Kirchlengern, Germany). The supernatant was transferred to a 5-mL volumetric flask, and the pellet was re-suspended with 2 mL of deionized water and centrifugated. Both supernatants were combined, and water was added to reach the desired volumes. The solution was passed through a 0.22-μm nylon syringe filter, and 20 μL of the sample was injected into an HPLC-DAD system (Dionex Ultimate 3000, Thermo Fisher Scientific, Waltham, MA, USA), equipped with a reverse-phase C18 column (PerfectSil^®^ ODS-3, 125 × 4.6 mm (4 μm), MZ Analysentechnik, Mainz, Germany). The mobile phases consisted of (A) 1:99 *v/v* acetic acid in water and (B) acetonitrile, eluted at 1 mL/min and 30 °C according to the following gradient: 0 min, 95% A; 8 min, 95% A; 20 min, 70% A; 25 min, 95 % A; 38 min, 95% A. The retention time of HMF was 5.5 min. The standard curve was prepared in 0.2 % *v/v* acetic acid in water. All analyses were performed in duplicate.

Before the analyses, the method was validated for linearity, specificity, accuracy, and precision [35]. Linearity was assessed in the range of 25–250 μg/L and expressed using the coefficient of determination (Table 2). LOD and LOQ were determined using the calibration curve, according to Equations (3) and (4).

#### 2.8.3. Heavy Metals

Total arsenic (As), cadmium (Cd), lead (Pb), mercury (Hg), and tin (Sn) were quantified in the dry seaweeds and in the extracts’ equivolumetric mixtures (see Section 2.7.1). Total As and Hg were analyzed using atomic absorption spectroscopy and Cd, Sn, and Pb using inductively coupled plasma atomic emission spectroscopy (ICP-AES). Dry seaweeds were processed using the TMECC 04.14 and 04.12-B methods [37] (LOQ (mg/kg): As, 1.50; Cd, 2.00; Pb, 1.00; Hg, 0.20; Sn, 2.00) and results were expressed as mg of metal per kg of dry weight. Extracts were analyzed according to the methods ME-12 [38], SM 3030 C and E, and 3120 B [39] (LOQ (mg/L): As, 0.006; Cd, 0.0015; Pb, 0.003; Hg, 0.001; Sn, 0.01), and results were expressed as mg of metal per kg of dry weight. Analyses were performed in duplicate.

### 2.9. Statistical Analysis

Total polyphenol content, antioxidant capacity, and mannitol analysis results are presented as means ±SD of five replicates (*n* = 5) per species, area, and thallus part combination. Carbohydrate-hydrolyzing enzyme activity and heavy metal assay results came from equivolumetric mixtures of the five replicates per sample group. STATGRAPHICS Centurion XVI software (Statgraphics Technologies, The Plains, VA, USA) was used to compare means through the analysis of variance (ANOVA) and Tukey post hoc tests. Correlations between variables were assessed using the Pearson correlation coefficient and the Steiger z-test. A 95% confidence level (α = 0.05) was used to determine the statistical significance in all analyses.

## 3. Results and Discussion

### 3.1. Effectivity of PHLE with 15% v/v Glycerol-Water against Maceration with 60% Acetone

Glycerol is a low-cost, food-grade, and environmentally friendly (bioderived, recyclable, and degradable) solvent [25]. Glycerol-water mixtures have been proven quite effective in extracting polyphenols from different plant matrices [22,25,26]. Therefore, we used a previously optimized glycerol-water PHLE process (15% *v/v*, 150 °C, 1500 psi) [26] to extract phlorotannins from *D. incurvata* and *L. spicata*. Maceration with 60% acetone (30 °C, 1 h) was used as the reference method because of its capacity to achieve high polyphenol yields from seaweeds [10,18].

In terms of TPC, the PHLE efficiencies (TPC _PHLE extracts_/TPC _acetone extracts_) varied according to the seaweed part used, showing a similar trend in both species: frond > holdfast > stipe (Figure 1a). The frond of *D. incurvata* presented a PHLE-TPC efficiency close to 180%, whereas that of *L. spicata* fronds was close to 130%. Both holdfasts showed a PHLE-TPC efficiency close to 100%, whereas that of the stipes were about 60%. A similar tendency was shown in the PHLE-AC efficiencies, measured as DPPH RSA (Figure 1b). The discrepancies among anatomical parts could be attributed to the solvents’ capacity to differentially extract distinct phlorotannin types according to their polarities and the unequal distribution of each phlorotannin type in the seaweed thallus. It has been proven that acetone increases the extraction efficiency of tannins associated with proteins because it can inhibit the phlorotannin–protein interaction [40].

The 15% *v/v* glycerol-water PHLE process can be considered an effective method to obtain phlorotannin extracts from the different thallus parts of *D. incurvata* and *L. spicata*, as it showed extraction efficiencies ranging from moderate to far above those of the comparison method, accomplished in a third of the time.

### 3.2. Effect of the Species, Harvesting Area, and Thallus Part Used on the TPC and AC of D. incurvata and L. spicata

This study’s goal was to investigate the conditions for the potential production of food-grade and high-phlorotannin-content extracts from *D. incurvata* and *L. spicata* using a previously optimized 15% *v/v* glycerol-water PHLE procedure. The influence of the species, harvesting area, and thallus part used on the TPC and AC of the extracts were explored.

As noted in Figure 2a, the factor that exerted the most significant influence on the TPC was the anatomical part used, with holdfasts showing the highest values (27.5 ± 6.3 mg PE/g dry seaweed). As expected, the same occurred for both AC measurements, DPPH RSA and ORAC (Figure 2b,c), with holdfasts showing mean values of 113 ± 25 and 163 ± 38 μmol TE/g dry seaweed, respectively. As verified by three-way ANOVA, the species also significantly affected TPC and DPPH RSA values, with *D. incurvata* showing the highest values. No statistically significant effect of harvesting area on TPC or AC was found; however, significant second-order interactions were detected (Appendix A).

DPPH RSA and ORAC presented the same trends. The correlation coefficients of both variables with TPC were high and not statistically different (0.96 vs. 0.97). DPPH RSA quantifies the antioxidants’ capacity to scavenge the artificial radical DPPH in organic media (e.g., methanol), whereas ORAC measures their ability to inhibit peroxyl radicals generated in aqueous media under physiological conditions. Thus, as peroxyl radicals are the predominant free radicals in biological systems, ORAC analysis is thought to be more suitable than DPPH RSA for determining the AC of compounds for pharmaceutical or food applications [41]. Another disadvantage of the DPPH RSA assay is related to the structure of DPPH; since the DPPH molecule is sterically hindered, high-molecular-weight compounds, such as phlorotannins, do not have full access to the radical site. Moreover, other compounds can impede the antioxidant from reaching the DPPH molecule [42].

In summary, regardless of the species (*D. incurvata* and *L. spicata*) and area (Las Cruces and Niebla), the holdfast showed the highest TPC and AC values. Hence, according to bio-ecological recommendations for a sustainable seaweed fishery, the entire alga’s utilization, including the holdfast, is suggested in order to recover the most phlorotannins from these species [43].

### 3.3. Extraction Yield, TPC, and AC of the PHLE Extract

TPC and AC were also expressed on a dry extract basis to determine the *D. incurvata* and *L. spicata* PHLE extracts’ technological value (Table 3). The extracts’ total solids were quantified, and results are presented as extraction yield values (% *w/w*, g extracted solids/100 g dry seaweed). Extraction yields were similar among the twelve groups, with only DLH and LNF being significantly different, with 36.8% ± 6.6% and 22.9% ± 3.7%, respectively. Yields in a similar range (20.4%–36.9%) were obtained with an optimized ultrasound-assisted extraction (UAE) procedure (50% *v/v* ethanol-water, 35 kHz, 30 min, 1:10 *w/v*) applied in ten Phaeophyta species. When the same samples were extracted with 50% ethanol-water using a conventional method (200 rpm, 20 °C, 4 h, 2 cycles, 1:15 *w/v*), yields were 50% smaller [44]. Using a UAE treatment similar to that of Ummat et al. (2020) [44], Agregán et al. (2018) [45] reported high yields in *Ascophyllum nodosum* and *Bifurcaria bifurcata* extracts (25.9% and 35.9%, respectively). Lower yields (15.5%–31.6%) were obtained using PHLE (95% *v/v* ethanol-water, 1500 psi, 160 °C, 20 min) to extract phlorotannins from *S. muticum* collected on the North Atlantic coasts [27]. Applying conventional methods (200 rpm, 24 h, 1:25 *w/v*), extraction yields from *F. vesiculosus* of 12.0% with 80% ethyl acetate at room temperature and 37.8% with water at 70 °C were obtained [10].

The values of TPC and AC expressed in terms of the extracts’ dry weight presented a variability similar to TPC and AC expressed as dry seaweed. Holdfast extracts showed the highest TPCs (mean ± SD: 95 ± 24 mg PE/g dry extract), with values ranging from 75 ± 14 to 129 ± 30 mg PE/g dry extract. Fronds and stipes ranged from 11.7 ± 1.8 to 41 ± 21 mg PE/g dry extract. Holdfast extracts’ TPCs were similar to those previously quantified in 95% ethanol-water PHLE extracts of *S. muticum* (75.43–148.97 mg PE/g dry extract or 1.02–4.70 g PE/100 g dry seaweed) [27]. Nevertheless, they were much lower (2–5-fold) than those systematically reported for other brown seaweed extracts, such as *F. vesiculosus* and *A. nodosum* acetone- or ethanol-water extracts prepared using maceration [10,46] or UAE [44,45]. *D. antarctica* and *L. spicata* extracts analyzed in previous studies presented TPC values similar to or lower than those obtained in this study. Conventional 50% ethanol-water extracts from *D. antarctica* and *L. spicata* collected in summer showed TPCs of 0.5 and 0.1 g PE/100 g dry seaweed [47], whereas 70% acetone-water extracts from fronds and stipes of *L. spicata* harvested in spring showed values of 28.02 and 40.20 mg PE/g dry extract [48].

Holdfast extracts showed the highest values in both AC assays; DPPH RSA values ranged from 238 ± 50 to 580 ± 180 μmol TE/g dry extract (mean ± SD: 400 ± 140 μmol TE/g dry extract) and ORAC values from 416 ± 73 to 740 ± 190 μmol TE/g dry extract (mean ± SD: 560 ± 130 μmol TE/g dry extract). Frond and stipe extract AC values were much lower; DPPH RSA results ranged from <LOQ to 160 ± 120 μmol TE/g dry extract and ORAC values from 126 ± 36 to 220 ± 110 μmol TE/g dry extract. The DPPH activities of the holdfast extracts described here were comparable to those of 50% ethanol-water UAE extracts from high-polyphenol-content Phaeophytas species, such as *F. vesiculosus*, *A. nodosum*, and *B. bifurcata*, which showed values in the range of 100–400 μmol TE/g dry extract [44,45]. However, their ORAC values were lower than those described for UAE extracts (50% ethanol-water), ranging from 1100 to 1600 μmol TE/g dry extract [45], and those reported for conventional methanolic extracts, which showed values between 1200 and 4200 μmol TE/g dry extract [49]. Our results were comparable to conventional 50% ethanol-water extracts obtained from brown seaweeds from the Northwest Mexican Pacific coast (190–820 μmol TE/g dry extract) [50]. All PHLE extracts characterized here presented higher ORAC values than those previously reported for *D. antarctica* and *L. spicata* 50% ethanol-water extracts [47].

The *D. incurvata* and *L. spicata* PHLE extracts presented similar yields and DPPH RSA values but lower TPCs than food-grade extracts of some well-studied Fucales species. Moreover, they showed comparable or higher TPC and AC values than those previously reported for *D. antarctica* and *L. spicata* conventional extracts. It is worth pointing out that direct comparisons among species or studies are not entirely accurate. Differences in the extraction conditions, the harvesting area and season, and other environmental factors to which the seaweeds were exposed could affect the polyphenol content and the antioxidant performance of the extracts.

### 3.4. Carbohydrate-Hydrolyzing Enzymes’ Inhibitory Activity

Postprandial hyperglycemia is an essential factor in developing insulin resistance and cardiovascular events in type 2 diabetes [51]. The inhibition of carbohydrate-hydrolyzing enzymes—α-glucosidase and α-amylase—is critical in reducing postprandial glycemia and preventing diabetes or diabetes complications [52,53]. Brown seaweed extracts have shown the capacity to reduce the activity of α-glucosidase or both enzymes at the same level as, or even more than, commercial hypoglycemic drugs (e.g., acarbose and miglitol) [18,53]. The α-glucosidase and α-amylase inhibition capacity of the PHLE extracts were assessed. Holdfast extracts were analyzed in the range of 10–260 μg/mL and frond and stipe extracts from 150 to 4000 μg/mL. 

For α-glucosidase, all holdfast extracts (Figure 3a) and three extracts from fronds and stipes (Figure 3b,c) displayed an inhibitory activity above 50% in the assayed ranges. LNH, LLH, and DNH extracts almost completely inhibited the enzyme from around 160 μg/mL, whereas LLS and DNF inhibited the enzyme from about 1000 μg/mL. As seen in Table 4, and according to the one-way ANOVA and Tukey post hoc tests (Appendix A), the DNH extract presented the highest α-glucosidase inhibitory capacity, with an IC_50_ value of 45.2 ± 1.6 μg/mL, i.e., around fifteen-fold lower than that of acarbose (659.5 ± 36.7 μg/mL). Except for DNS, the other active extracts showed IC_50_ values far lower than acarbose, ranging from 62.6 ± 2.4 to 324.1 ± 6.5 μg/mL. These inhibition results positively correlate with the TPC, DPPH RSA, and ORAC values of the extracts, showing correlation coefficients of 0.90, 0.93, and 0.87, respectively (not statistically different). It seems that holdfast polyphenols are more bioactive than stipe and frond polyphenols since the differences between holdfast and stipe/frond extracts are more significant in glucosidase inhibition activity than in TPC and AC.

Different α-glucosidase inhibitory capacity results have been found in other brown seaweed species. For instance, *F. vesiculosus* acetone-water extracts have shown IC_50_ values ranging from eight-fold lower than acarbose (32 ± 3 vs. 264 ± 41 μg/mL) to two thousand-fold lower than acarbose (0.34 ± 0.01 vs. 720 ± 10 μg/mL) [18,54,55]. Ethanolic and acetone-water *A. nodosum* extracts presented IC_50_ values ranging between twenty and two thousand-fold lower than acarbose [54]. Although the ORAC values found in Mexican seaweed ethanolic extracts were comparable to some of the PHLE extracts, their α-glucosidase inhibition capacities were far lower (one to four-fold lower than the acarbose IC_50_) [50]. Previous studies with *D. antarctica* and *Lessonia* extracts have presented lower α-glucosidase inhibitory activities than those found in this research. Pacheco et al. (2020) [34] reported IC_50_ values two times lower and seven times higher than acarbose, respectively, for southern Chilean *D. antarctica*’s and *L. spicata*’s PHLE ethanol-water extracts (473 ± 0.9 and 5318 ± 0.8 μg/mL vs. 798 ± 1.1 μg/mL). Yuan et al. (2018) [56] found IC_50_ values to be four times lower than those of acarbose for methanol-water extracts from *Lessonia trabeculata* and observed no α-glucosidase inhibitory activity for similar *L. nigrescens* extracts.

Regarding the α-amylase inhibitory capacity, all PHLE extracts proved to be inactive at the assayed concentrations (Figure 4), whereas acarbose showed an IC_50_ of 75.5 ± 4.7 μg/mL. These results are consistent with previous studies [34]. Even though some authors have reported brown seaweed extracts with similar inhibitory effects to those of acarbose, such as cold water extracts of *A. nodosum* and *F. vesiculosus* [53], in general, they have demonstrated low α-amylase inhibitory activity [18,50,57]. This behavior might be beneficial in terms of glycemic homeostasis. It has been observed that the initial starch digestion carried out in the mouth by α-amylase triggers a preabsorptive insulin release (cephalic phase), preparing the body for the upcoming glucose [58].

Overall, the *D. incurvata* and *L. spicata* food-grade PHLE extracts, especially those from the holdfast, constitute a promising alternative to traditional α-glucosidase inhibitors. Similar extracts from the same species demonstrated no cytotoxic effects in human colon adenocarcinoma HT-29 cells at the inhibitory concentrations reported here and up to 1000 μg/mL [34]. Although the α-glucosidase inhibitory capacity has been mainly correlated with phlorotannins, some authors have found the same capacity in fucoxanthin- and fucoidan-rich brown seaweed extracts [18,59,60]. Therefore, further characterization analyses or purification steps must be performed in *D. incurvata* and *L. spicata* PHLE extracts to verify the association between α-glucosidase inhibition and the extracts’ phlorotannin content.

### 3.5. Presence of Contaminants: Mannitol, HMF, and Heavy Metals

Crude extracts contain desired metabolites and co-extracted matrix components that could affect their applicability and safety as food or pharmaceutical ingredients. Significant glucose and fructose concentrations have been detected in high-polyphenol-content PHLE extracts from grape pomace [22]. Additionally, high extraction temperatures may produce potential carcinogens derived from non-enzymatic browning reactions, such as HMF, limiting the extracts’ applicability [61]. Therefore, to evaluate the relevance of seaweed PHLE extracts as potential nutraceuticals or functional food ingredients, we quantified mannitol, heavy metals, and HMF.

#### 3.5.1. Mannitol and HMF in PHLE Extracts

Carbohydrates are the major components of brown seaweeds. They include fucoidan, laminarin, alginate, and mannitol. Given its relevance as an energy source, mannitol is one of the most abundant carbohydrates in seaweeds, accounting for 3–21%—even up to 30%—of the alga’s dry weight, depending on the species, season, and growth conditions [62]. The mannitol concentrations of the PHLE extracts, expressed as dry seaweed and dry extract, are presented in Table 5. According to a three-way ANOVA and Tukey post hoc tests (Appendix A), *D. incurvata* showed significantly higher mannitol contents than *L. spicata* (means ± SD: 11.5% ± 3.5% vs. 5.6% ± 1.3%), and the Las Cruces samples contained more mannitol than the Nieblas samples (means ± SD: 10.0% ± 4.8% vs. 7.2% ± 2.5%). In general, mannitol contents ranged from 3.6% ± 0.8% to 16.4% ± 3.0% of dry seaweed, similar to those reported previously in brown algae [63,64]. Mannitol concentrations expressed in a dry extract basis ranged from 147 ± 42 to 473 ± 59 mg/g, that is, up to 50% of the extract weight. These large concentrations are relevant, as mannitol has been found to produce abdominal symptoms in people with irritable bowel syndrome [65]. Otherwise, mannitol has been described as a free radical scavenger [66], interfering with TPC and AC measurements.

HMF has been reported as a by-product of PHLE processes performed at high temperatures (over 120 °C) in various plant materials [24]. As HMF has been recognized as a cytotoxic compound and a potential carcinogen for humans [67], its content in the PHLE extracts was assessed. No detectable concentration of HMF was found in any extract (LOD: 0.012 mg/L). The generation of HMF has been commonly described as a side reaction of the thermal acid hydrolysis of seaweeds (e.g., 121 °C, 0.5 M HCl, 15 min), a typical pretreatment for bioethanol production [68,69]. Mannitol is not a good HMF precursor, as its hydrothermal decomposition (0.5 % *w/w*, 10 min, 5 MPa) produced low levels of HMF (up to 5 μM) at temperatures under 220 °C [70]. The lack of HMF in the extracts could also be attributed to mannitol’s capacity to form adducts with specific amino acids [71].

#### 3.5.2. Heavy Metals in Dry Seaweeds and PHLE Extracts

The world’s per capita intake of algae is continuously growing, raising concerns about the hazardous concentrations of toxic metals they may contain [7,72]. Concentrations of Cd and As above the maximum permitted levels have been identified in several edible seaweeds and seaweed-derived products [73,74]. In general, brown algae have a higher biosorption capacity than red and green algae; in fact, many countries recommend avoiding the consumption of the brown seaweed Hijiki (*Sargassum fusiforme*) due to its potentially dangerous levels of inorganic As, the toxic form of this metalloid [7]. Only a few countries regulate the maximum levels of heavy metals in algae and derived foods. The French law is the most complete in this regard, establishing limits for inorganic As (3 mg/kg dw), Cd (0.5 mg/kg dw), Pb (5 mg/kg dw), Hg (0.1 mg/kg dw), and Sn (5 mg/kg dw) for edible seaweeds [75]. The content of these heavy metals in the seaweed samples and the PHLE extracts are presented in Table 6.

The concentrations of total As in the seaweed samples ranged from 8.29 to 84.08 mg/kg dw. According to a three-way ANOVA and Tukey post hoc tests (Appendix A), holdfasts showed higher levels than stipes and fronds (means ± SD: 49.7 ± 25.4 mg/kg dw vs. 16.7 ± 9.6 and 21.2 ± 8.2 mg/kg dw), and samples of *L. spicata* from Niebla presented higher concentrations than their counterparts. Differences in the As content among alga parts and locations were also seen in the Phaeophyta *Alaria nana* (Alaska), with holdfasts showing the highest levels [76]. Arsenic values described here were in the range reported for other brown seaweeds, including species from *Laminaria* and *Sargassum* genus and *Eisenia bicyclis* (arame) [73,75,77,78]. However, the As concentrations in fronds of this study were smaller. The arsenic content of *D. incurvata* and *L. spicata* fronds (means ± SD: 7.9 ± 4.9 and 27.8 ± 2.1 mg/kg dw) were lower than those previously reported for *D. antarctica* and *L. nigrescens* (49.0 ± 34.5 and 57.1 ± 22.8 mg/kg dw) [79]. Although inorganic As (_i_AS) was not assessed in this study, it is known that the _i_AS content of brown seaweeds is not greater than 10% (and generally < 2%) of the total As, with the exception of *S. fusiforme*, in which the _i_AS level can reach up to 80% of the total As [74,75,76]. _i_AS concentrations representing 0.6% of the total As (0.31 and 0.35 mg/kg dw) were found in the abovementioned *D. antarctica* and *L. nigrescens* samples [79]. Therefore, the _i_AS contents of the studied species *D. incurvata* and *L. spicata* would probably not exceed the maximum limits established in some countries (France/USA: 3 mg/kg dw, Australia/New Zealand: 1 mg/kg dw) [80], although this must be further confirmed. On the other hand, the total As concentrations of derived extracts ranged between 14.88 and 258.47 mg/kg dw. These higher values mean that all or almost all the seaweed’s As content was extracted by the PHLE process. Thus, the dry PHLE extracts are more likely to have _i_As levels close to or even above the maximum acceptable limits.

The cadmium concentrations in the seaweed samples ranged from 0.84 to 4.81 mg/kg dw, and no clear trend could be observed (mean ± SD: 2.2 ± 1.2 mg/kg dw), although *L. spicata*’s values were slightly higher than *D. incurvata*’s. These Cd contents were higher than those found in the *Laminaria* species *E. byciclis* and *Undaria pinnatifida* (wakame) from Europe; and were at the same level or even lower than those described for the *Sargassum* species *U. pinnatifida* from Asia and *Fucus spiralis* [72,73,75,77,78,81]. Sáez et al. (2012) [82] reported high Cd concentrations (0.5–3.5 mg/kg dw) for the different thallus parts of *L. trabeculata* harvested in Chile; the holdfast showed higher concentrations than the stipe and the latter showed higher values than the frond. A Cd level of 2.46 mg/kg dw was detected in a Chilean *D. antarctica* sample, which is higher than the mean of the *D. incurvata* samples in this study (1.3 ± 0.5 mg/kg dw) [7]. All samples reported here greatly exceeded the maximum permitted concentrations established by French and Australian/New Zealand regulations for edible seaweeds (0.5 and 0.2 mg/kg dw) [73]. Hence, their consumption could represent a risk for human health in terms of Cd intake. The contribution of the edible seaweed *D. incurvata* to the total daily intake (TDI) of Cd must be determined, especially considering that other marine products and some vegetables are also Cd accumulators [83,84]. In the other PHLE extracts, Cd concentrations ranged between 0.20 and 6.13 mg/kg dw, with only holdfast extracts showing values over 0.38 mg/kg dw. The Cd values of frond and stipe extracts were lower than those of the dry seaweed parts, indicating that the PHLE process extracted just a small amount of Cd from the source matrix. Concerning the EU regulations for seaweed-derived foods, 25% of the extracts exceeded the limit established for “food supplements consisting exclusively or mainly of seaweed, or products derived from seaweed” (3 mg/kg ww) [85]. Cd concentrations well below the EU and French limits but above the Australian regulations were found in tablets and soluble concentrates of *Fucus* species (0.011–0.356 mg/kg dw) [73].

The concentration of Pb was below the LOQ of the analytical method (1.0 mg/kg dw) in all seaweed samples. Pb levels in brown seaweeds are usually under 1.0 mg/kg, although higher values have been detected in *Sargassum* species and *U. pinnatifida* [73,75,77]. In contrast to As and Cd, Pb values found in samples from the Norwegian coasts were lower in brown seaweeds than in red and green seaweeds [74]. Sáez et al. (2012) [82] reported values greater than 4 mg/kg dw in fronds of *L. trabeculata* harvested from Pb-polluted seawaters in Chile; however, Pb concentrations were below 1 mg/kg in stipes and holdfasts. No Pb content was detected in a Chilean *D. antarctica* sample [73]. All seaweed samples analyzed here presented Pb levels under the French limit (5 mg/kg dw), and therefore their consumption may not pose a risk for human health in terms of Pb toxicity. The Pb levels of the PHLE extracts ranged from 0.28 to 0.81 mg/kg dw. These low values mean that the PHLE process extracted just a small part of the Pb content from the seaweed matrix. The Pb content of all extracts was under the French limit for edible seaweeds. Pb concentrations well below (<LOD–0.25 mg/kg dw) and others much higher (11.3–14 mg/kg dw) than those of this study have been found in *Fucus* tablets [73].

Hg was not detected in any seaweed samples (LOD: 0.001 mg/kg dw) and therefore it was not analyzed in the extracts. In general, Hg contents in red, green, and brown algae are under 0.5 mg/kg dw. In brown seaweeds, concentrations below 0.01 mg/kg dw have been systematically reported for *Laminaria* and *Fucus* species, and slightly higher levels —0.03–0.05 mg/kg dw—have been found in *U. pinnatifida*, *S. fusiforme*, and *Pelvetia canaliculata* from Norway [74,75,78]. Higher than usual Hg concentrations (over 0.5 mg/kg dw) were reported by Squadrone et al. (2018) [86] in seaweed species from the Italian Mediterranean coasts. The low Hg contents of brown seaweeds should not be a concern according to the French legislation (limit: 0.1 mg/kg dw); however, special care must be taken considering the EU limit for edible seaweeds (0.01 mg/kg).

Concentrations of Sn in the seaweed samples ranged from not detected (LOD: 0.01 mg/kg dw) to the LOQ of the analytical method (2.00 mg/kg dw). On the other hand, Sn was not detected in the analyzed PHLE extracts. The literature about the Sn content in seaweeds is scarce because this metal is mainly associated with canned food [87]. Brown seaweeds from coastal areas of Italy showed levels under 1.0 mg/kg dw, except for *Halopteris filicina,* which presented an Sn content of 1.8 ± 0.04 mg/kg dw [86]. *Padina pavonica* and *Cystoseira mediterranea* collected in Sardinian coastal areas showed Sn values below 0.05 mg/kg dw [88]. Sn concentrations between 0.11–0.17 mg/kg dw were found in species belonging to the Chlorophyta *Ulva* genera grown in fishpond aquaculture systems [89]. Higher Sn levels (<1–34 mg/kg dw) were found in samples from edible seaweeds purchased in the Italian market [90]. The study reported levels close to or well above the French limit for Sn (5 mg/kg dw) in *A. nodosum* (4.7 ± 0.7 mg/kg dw) and *Laminaria digitata* (34.0 ± 5.1 mg/kg dw); interestingly, the concentrations of Cd and Pb were also above the available regulations in some seaweed samples.

In summary, potentially toxic Cd levels were found in *D. incurvata* and *L. spicata* from central and southern Chile, which is of concern since *D. incurvata* is the most consumed algae in Chile, and new *D. incurvata*-based foodstuffs are continuously emerging. The high As concentrations, also reported for both seaweeds, should not constitute a health risk due to the low _i_As (the toxic As species) contents systematically detected in most seaweeds, particularly in *D. incurvata*; however, this remains to be confirmed. Studies suggest that continuous monitoring of the heavy metal contents of seaweeds must be performed in order to protect consumers’ health, as the concentrations fluctuate according to environmental factors (e.g., anthropic activity, location, season, and pH). As, Cd, and Sn levels were even higher in some PHLE extracts than in dry seaweeds due to the concentration effect of the extraction process. Since some of those metal contents exceeded the available regulations, and mannitol levels were also high—accounting for up to 50% of the extract dry weight—further purification steps are suggested if food or pharmaceutical applications are intended for the PHLE extracts.

## 4. Conclusions

Through an effective, environmentally friendly, and food-grade PHLE process, *Durvillaea incurvata* and *Lessonia spicata* phlorotannin extracts with high antioxidant and antihyperglycemic potential were obtained. A glycerol-based PHLE process was used for the first time in the extraction of the bioactive compounds of seaweed. The anatomical part of the alga significantly impacted the dry seaweeds’ and extracts’ polyphenol content and antioxidant activity, with the holdfast showing the highest values. No α-amylase inhibitory activity was found. Instead, holdfast extracts presented α-glucosidase inhibitory capacities several-fold higher than that of acarbose, with *D. incurvata*’s holdfast from Niebla being the most active. Elevated concentrations of mannitol and potentially toxic cadmium levels were found in seaweeds and extracts. Therefore, the utilization of the whole plant, including the holdfast, and the implementation of further purification steps are recommended to use these seaweed PHLE extracts as effective and safe antihyperglycemic agents.

## Figures and Tables

**Figure 1 antioxidants-10-01105-f001:**
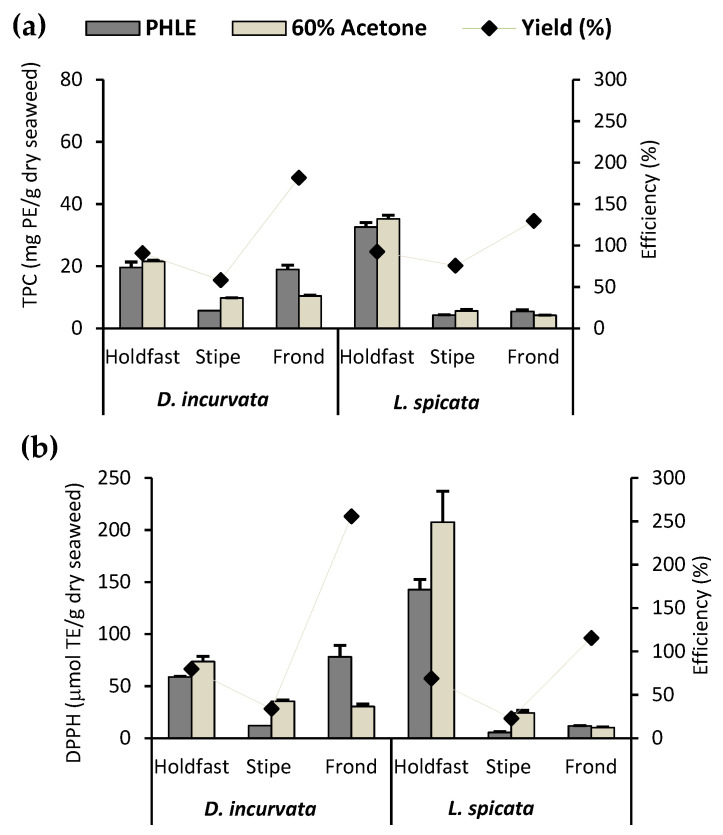
Effectivity of PHLE against 60% acetone in the extraction of phlorotannins from the main anatomical parts of *D. incurvata* and *L. spicata*: holdfast, stipe, and frond. (**a**) The TPC (mg PE/g dry seaweed) and (**b**) the antioxidant capacity, measured as DPPH RSA (μmol TE/g dry seaweed), were analyzed in PHLE and 60% acetone extracts obtained from a randomly selected sample per each species/thallus part combination. PHLE-TPC and PHLE-AC efficiencies were determined as a ratio of the 60% acetone extracts’ values. Bars represent the means ± SD of three extractions carried out per each sample and method, and dots correspond to the mean efficiencies of PHLE.

**Figure 2 antioxidants-10-01105-f002:**
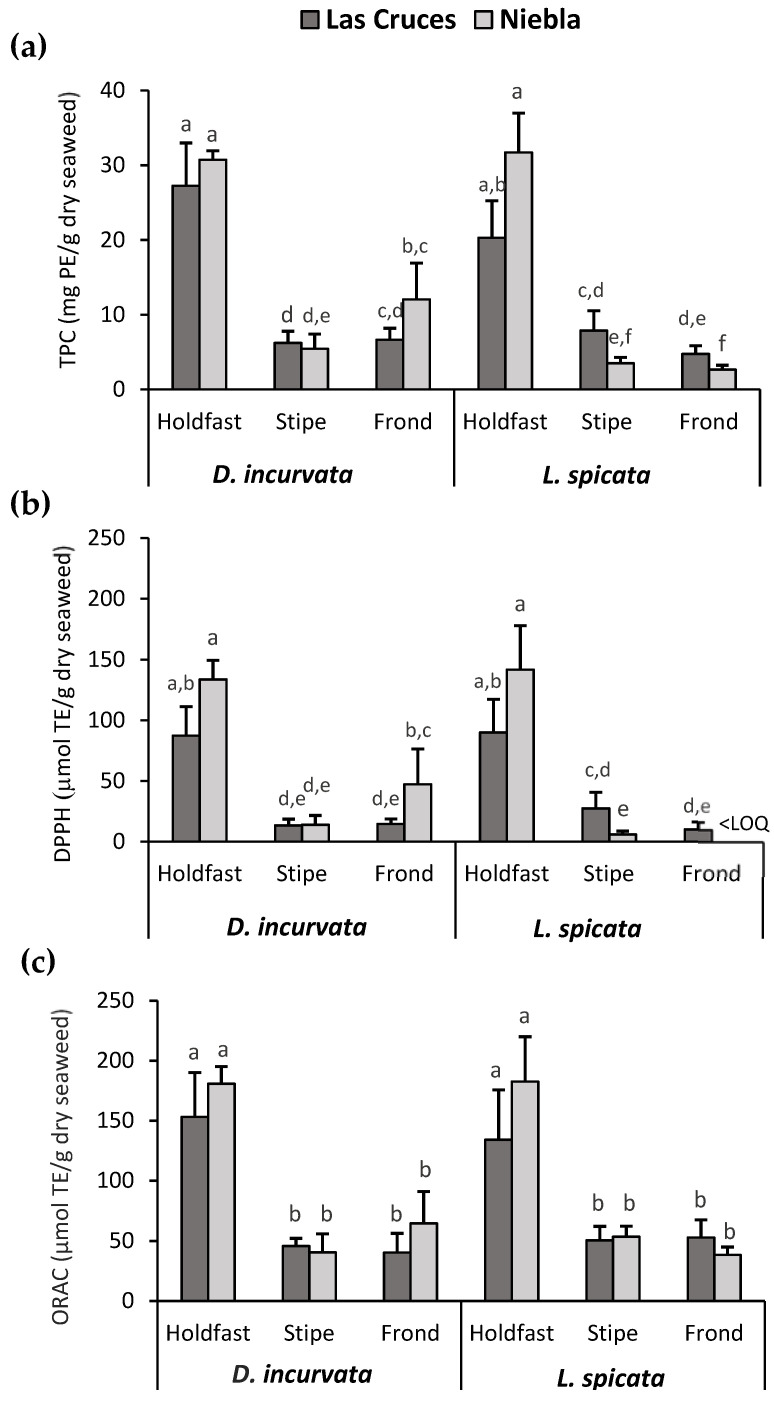
Spatial, anatomical, and interspecies variability of the phlorotannin content and antioxidant capacity of the seaweeds *D. incurvata* and *L. spicata* harvested in central (Las Cruces) and southern (Niebla) Chile. (**a**) The TPC (mg PE/g dry seaweed) and (**b**,**c**) the antioxidant capacity, quantified by DPPH RSA and ORAC (μmol TE/g dry seaweed), were determined in the PHLE extract. Bars represent the means ± SD of five replicates. Means were compared using one-way ANOVA and Tukey post hoc tests (α ≤ 0.05). Different letters indicate significantly different groups for each response. <LOQ: below the limit of quantification.

**Figure 3 antioxidants-10-01105-f003:**
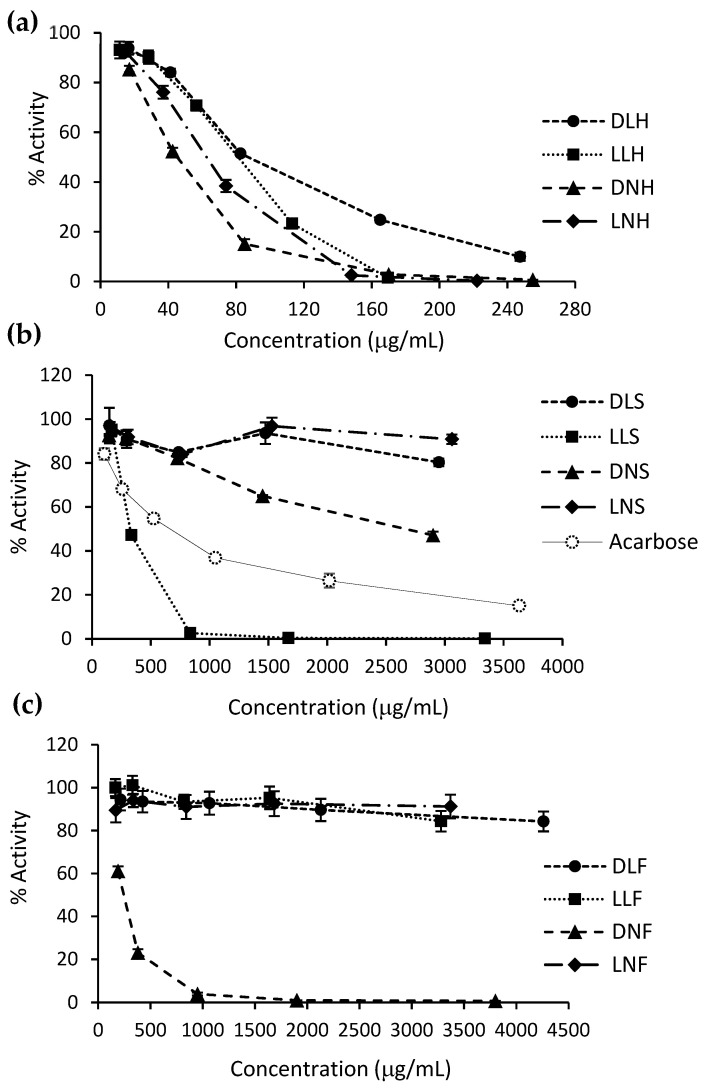
α-glucosidase inhibitory activity of *D. incurvata* and *L. spicata* PHLE extracts. The charts present the relative enzyme activity against increasing concentrations of extracts, classified by the seaweed part origin: (**a**) holdfast, (**b**) stipe, and (**c**) frond. The acarbose inhibitory curve is also shown. Data are means ± SD of two replicates.

**Figure 4 antioxidants-10-01105-f004:**
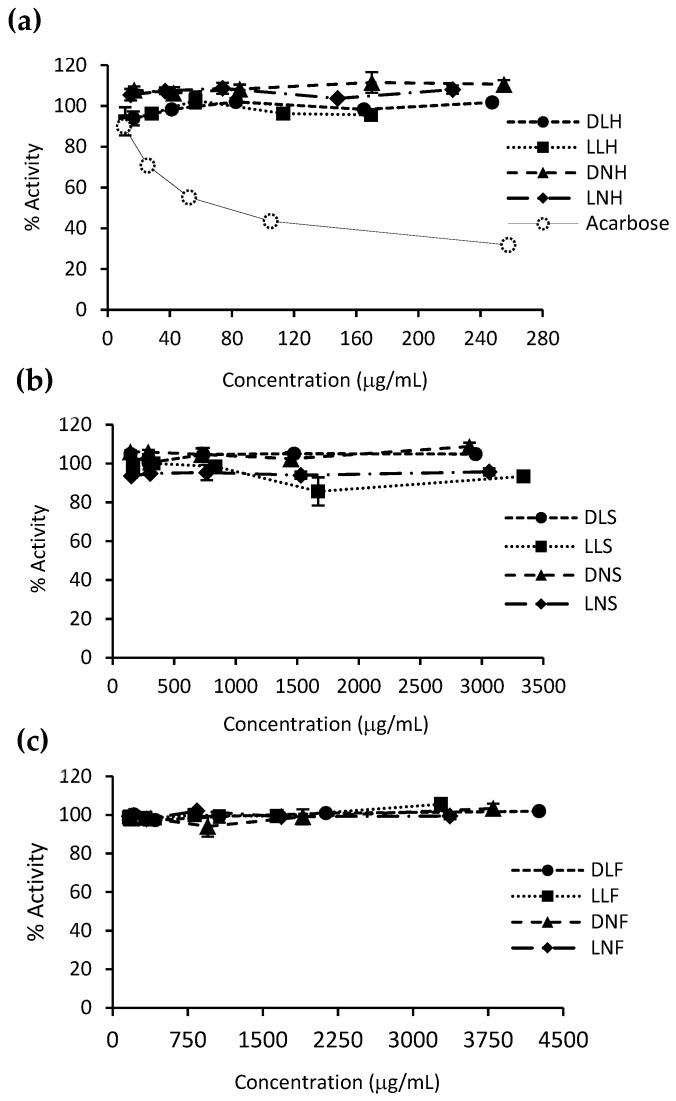
α-amylase inhibitory activity of *D. incurvata* and *L. spicata* PHLE extracts. The charts present the relative enzyme activity against increasing concentrations of the extracts, classified by the seaweed part origin: (**a**) holdfast, (**b**) stipe, and (**c**) frond. The acarbose inhibitory curve is also shown. Data are means ± SD of two replicates.

**Table 1 antioxidants-10-01105-t001:** Identification and description of the twelve sample groups assessed in this study. Each group includes five replicates.

Species	Area	Alga Part	Group ID
*D. incurvata*	Las Cruces	Holdfast	DLH
		Stipe	DLS
		Frond	DLF
	Niebla	Holdfast	DNH
		Stipe	DNS
		Frond	DNF
*L. spicata*	Las Cruces	Holdfast	LLH
		Stipe	LLS
		Frond	LLF
	Niebla	Holdfast	LNH
		Stipe	LNS
		Frond	LNF

**Table 2 antioxidants-10-01105-t002:** Validation parameters for mannitol and HMF.

Compound	Concentration Range (mg/L)	Slope	Interception	Coefficient of Determination (R^2^)	LOD (mg/L)	LOQ (mg/L)	RSD
Mannitol	100–1000	0.0031	−0.0399	0.9986	42	126	0.039
HMF	0.025–0.250	0.0042	−0.0332	0.9983	0.012	0.036	0.015

**Table 3 antioxidants-10-01105-t003:** Total polyphenol content, DPPH radical scavenging activity, and ORAC values of the twelve sample groups expressed on a dry extract basis. Extraction yield (% *w/w*, g extracted solids/100 g dry seaweed) and TPC denoted as g PE/100 g dry seaweed are also shown.

Group ID	Extraction Yield	TPC		DPPH RSA	ORAC
	(% Dry Seaweed)	(mg PE/g Dry Extract)	(g PE/100 g Dry Seaweed)	(μmol TE/g Dry Extract)	(μmol TE/g Dry Extract)
DLH	36.8 ± 6.6 ^a^	75 ± 14 ^a,b^	2.7 ± 0.6 ^a^	238 ± 50 ^a,b^	416 ± 73 ^a^
DLS	34.7 ± 5.0 ^a,b^	17.9 ± 3.4 ^d^	0.6 ± 0.2 ^d^	39 ± 14 ^d,e^	135 ± 32 ^b^
DLF	33.6 ± 6.7 ^a,b^	21.1 ± 9.2 ^c,d^	0.7 ± 0.2 ^c,d^	47 ± 23 ^d,e^	126 ± 36 ^b^
DNH	34.7 ± 4.8 ^a,b^	90 ± 14 ^a^	3.1 ± 0.1 ^a^	392 ± 71 ^a,b^	530 ± 100 ^a^
DNS	31 ± 11 ^a,b^	20 ± 11 ^c,d^	0.5 ± 0.2 ^d,e^	49 ± 25 ^c-e^	144 ± 63 ^b^
DNF	31.5 ± 7.1 ^a,b^	41 ± 21 ^b,c^	1.2 ± 0.5 ^b,c^	160 ± 120 ^b,c^	220 ± 110 ^b^
LLH	23.8 ± 5.2 ^a,b^	85 ± 12 ^a^	2.0 ± 0.5 ^a,b^	377 ± 92 ^a,b^	559 ± 96 ^a^
LLS	34.4 ± 7.0 ^a,b^	24 ± 10 ^c,d^	0.8 ± 0.3 ^c,d^	82 ± 45 ^c,d^	152 ± 45 ^b^
LLF	30.3 ± 3.0 ^a,b^	16.3 ± 5.4 ^d^	0.5 ± 0.1 ^d,e^	35 ± 24 ^d,e^	181 ± 62 ^b^
LNH	24.9 ± 2.6 ^a,b^	129 ± 30 ^a^	3.2 ± 0.5 ^a^	580 ± 180 ^a^	740 ± 190 ^a^
LNS	27.0 ± 8.0 ^a,b^	13.6 ± 3.9 ^d^	0.3 ± 0.1 ^e,f^	22 ± 8 ^e^	212 ± 69 ^b^
LNF	22.9 ± 3.7 ^b^	11.7 ± 1.8 ^d^	0.3 ± 0.1 ^f^	<LOQ	174 ± 51 ^b^

Data are means ± SD (*n* = 5). Means were compared using one-way ANOVA and Tukey post hoc test (α ≤ 0.05). Different letters indicate significantly different groups for each response.

**Table 4 antioxidants-10-01105-t004:** IC_50_ values of the active seaweed PHLE extracts against α-glucosidase at the assayed ranges.

Group ID	IC50 (μg/mL) α-Glucosidase
DLH	87.1 ± 0.8 ^c^
DNH	45.2 ± 1.6^a^
DNS	2700 ± 100^g^
DNF	245.1 ± 5.3^d^
LLH	81.2 ± 0.9^c^
LLS	324.1 ± 6.5^e^
LNH	62.6 ± 2.4^b^
Acarbose	659.5 ± 36.7^f^

Data are means ± SD of two replicates. Different letters indicate significantly different groups (α ≤ 0.05).

**Table 5 antioxidants-10-01105-t005:** Mannitol content of seaweed PHLE extracts expressed as a percentage of dry seaweed and on a dry extract basis. Data are means ± SD (*n* = 5).

Group ID	Mannitol	
	(% Dry Seaweed)	(mg/g Dry Extract)
DLH	15.3 ± 2.7	423 ± 85
DLS	16.4 ± 2.9	473 ± 59
DLF	10.0 ± 1.8	310 ± 120
DNH	10.4 ± 0.9	303 ± 46
DNS	9.3 ± 2.6	360 ± 240
DNF	7.7 ± 1.4	248 ± 39
LLH	5.9 ± 1.5	260 ± 110
LLS	7.2 ± 1.4	213 ± 43
LLF	5.0 ± 1.4	168 ± 46
LNH	3.6 ± 0.8	147 ± 42
LNS	6.6 ± 1.0	257 ± 56
LNF	5.4 ± 1.5	237 ± 46

**Table 6 antioxidants-10-01105-t006:** Heavy metal content (Total As, Cd, Pb, Hg, and Sn) of seaweed samples (S) and derived PHLE extracts (E) expressed as mg/kg of dry weight. Data are means of two replicates.

Group ID	Total As		Cd		Pb		Hg		Sn	
	S	E	S	E	S	E	S	E	S	E
DLH	23.06	49.09	1.38	0.27	<1.00	0.55	ND	n.a.	<2.00	ND
DLS	10.83	20.11	1.70	0.22	<1.00	0.43	ND	n.a.	<2.00	ND
DLF	17.80	27.96	0.84	0.22	<1.00	0.44	ND	n.a.	<2.00	ND
DNH	48.62	90.88	2.15	3.71	<1.00	0.28	ND	n.a.	ND	n.a.
DNS	8.29	14.88	1.07	0.20	<1.00	0.41	ND	n.a.	ND	n.a.
DNF	11.30	26.58	0.86	0.23	<1.00	0.47	ND	n.a.	ND	n.a.
LLH	43.09	202.19	3.13	6.13	<1.00	0.46	ND	n.a.	<2.00	ND
LLS	17.81	39.53	2.33	0.28	<1.00	0.56	ND	n.a.	<2.00	ND
LLF	26.28	55.40	4.51	0.31	<1.00	0.62	ND	n.a.	ND	n.a.
LNH	84.08	258.47	4.35	5.38	<1.00	0.81	ND	n.a.	<2.00	ND
LNS	29.78	61.81	1.87	0.34	<1.00	0.69	ND	n.a.	ND	n.a.
LNF	29.27	35.68	1.57	0.38	<1.00	0.75	ND	n.a.	<2.00	ND

ND, not detected. n.a., not analyzed.

## Data Availability

Data are contained within the article and Appendix A.

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
