# Peer review of "Pressurized Hot Liquid Extraction with 15% v/v Glycerol-Water as An Effective Environment-Friendly Process to Obtain Durvillaea incurvata and Lessonia spicata Phlorotannin Extracts with Antioxidant and Antihyperglycemic Potential"

_antioxidants, 2021, doi:10.3390/antiox10071105_

Round 1

Reviewer 1 Report

Manuscript ID: antioxidants-1273359

Title: Pressurized Hot Liquid Extraction with 15% v/v glycerol-water as an effective environment-friendly process to obtain Durvillaea incurvata and Lessonia spicata phlorotannin extracts with antihyperglycemic potential

The manuscript describes the investigation of two brown seaweeds (Durvillaea incurvata and Lessonia spicata) for obtaining phlorotannin extracts with antihyperglycemic potential. The authors proposed the utilization of Pressurized Hot Liquid Extraction as extraction method, by using an environment-friendly solvent. They studied the influence of a few factors (the species, harvesting area and anatomical part) on the extracts composition and properties (polyphenol content, antioxidant capacity, carbohydrate-hydrolyzing enzymes, inhibitory activity, contaminants). In conclusion, the authors recommended the utilization of the whole brown seaweeds and the implementation of further purification steps for using the seaweed PHLE extracts as effective and safe antihyperglycemic agent.

In my opinion, the manuscript is well written and comprehensively documented. The authors have done a large number of experimental determinations and their results may be useful to other researchers working in this field. I recommend the publication of the manuscript.

Author Response

Thank you very much for your encouraging words.

Reviewer 2 Report

The manuscript by Erpel, et al. presences a study that compares the extracts of different parts of brown seaweed. The experiments are explained in detail and serval aspects of the extracts are examined. I recommend that the manuscript be accepted after minor corrections.

The title, keywords, abstract and conclusion focus on the 15 % glycerol extraction, however, from the manuscript I do not see any novelty in this extraction procedure. The extract procedure was used here without optimization and compared to the simpler acetone, maceration it does not perform better. I do see why an extraction which requires instrumentation for higher pressure and temperature would be used in any future study. To me, the novelty of the work is the identification of hazards such as Cd while quantifying significant amounts of phlorotannins. The title, keywords, abstract and conclusion should be rewritten to focus more on this and not the extraction procedure.

Is there an explanation for why LLS has a much stronger effect on activity than the other samples (Figure 3b).

Author Response

Thank you very much for your recommendations.

Even though some previous studies reported PHLE as an environmentally friendly process to obtain seaweed’s bioactive compounds, all of them employed water or ethanol-water mixtures as the extraction solvent (Article references.: 19,20). To the best of our knowledge, this is the first study testing a glycerol-water mixture in PHLE to obtain phlorotannin extracts, and therefore we think this is a relevant aspect of our research. Additionally, performing process optimization was not our goal considering the large number of samples to analyze. Instead, we used previously optimized conditions (Art. Ref.: 22,26). We verified that the 15% v/v glycerol-water PHLE process is effective, compared with a non-environmentally friendly process (maceration with acetone-water), in the obtention of phlorotannin extracts with antioxidant and antihyperglycemic potential from seaweeds. Considering the assessment of different factors —i.e., species, alga part, and zone— we concluded that the holdfast extracts presented the highest bioactivities. We now expect to optimize the extraction conditions (temperature, time, cycles, % glycerol) to maximize the recovery of those phlorotannins associated with the highest antihyperglycemic potential.

This study also intends to alert about the potential hazardous heavy metal concentrations, mainly cadmium, that seaweed and seaweed-derived food may have. Measurement of heavy metals in the seaweed extracts has been poorly covered in previous research,

Answering the question “Is there an explanation for why LLS has a much stronger effect on activity than the other samples? (Figure 3b)”, we can explain the much stronger activity of LLS by the fact that this sample has the highest Total Polyphenol Content (mg PE/g dry extract) and antioxidant capacity (μmol TE/g dry extract, measured as DPPH). 

Reviewer 3 Report

Please find attached my review letter.

Author Response

We intended to analyze our non-purified extracts with HPLC; however, their high contaminant content (e.g., mannitol) impeded a reliable identification of phlorotannins. Highly concentrated samples were not injected due to equipment limitations. We are now performing HPLC-MS and MS/MS analyzes to identify the phlorotannin structures present in the PHLE purified extracts. Phlorotannins with a high polymerization degree —of up to 20 phloroglucinol units (PGU)— were detected in the most active extracts (e.g., DNH, DLH DNF).

In our study, we have identified the most promising matrices to obtain phlorotannin extracts with the highest antihyperglycemic potential. We expect to associate specific phlorotannins with this bioactivity (using purified extracts) and then optimize the extraction conditions (temperature, time, cycles, % glycerol) to maximize their recovery.

Our native English editor had checked the final text.